# Geometrical Aspects in the Analysis of Microcanonical Phase-Transitions

**DOI:** 10.3390/e22040380

**Published:** 2020-03-26

**Authors:** Ghofrane Bel-Hadj-Aissa, Matteo Gori, Vittorio Penna, Giulio Pettini, Roberto Franzosi

**Affiliations:** 1Dipartimento di Scienze fisiche, della Terra e dell’ambiente (DSFTA), University of Siena, Via Roma 56, 53100 Siena, Italy; 2Quantum Biology Lab, Howard University, 2400 6th St NW, Washington, DC 20059, USA; 3Dipartimento di Fisica, Politecnico di Torino, Corso Duca degli Abruzzi 24, I-10129 Torino, Italy; 4Dipartimento di Fisica Università di Firenze, and I.N.F.N., Sezione di Firenze, via G. Sansone 1, I-50019 Sesto Fiorentino, Italy; 5QSTAR & CNR—Istituto Nazionale di Ottica, Largo Enrico Fermi 2, I-50125 Firenze, Italy

**Keywords:** microcanonical ensemble, phase transitions, differential geometry

## Abstract

In the present work, we discuss how the functional form of thermodynamic observables can be deduced from the geometric properties of subsets of phase space. The geometric quantities taken into account are mainly extrinsic curvatures of the energy level sets of the Hamiltonian of a system under investigation. In particular, it turns out that peculiar behaviours of thermodynamic observables at a phase transition point are rooted in more fundamental changes of the geometry of the energy level sets in phase space. More specifically, we discuss how microcanonical and geometrical descriptions of phase-transitions are shaped in the special case of $\phi^{4}$ models with either nearest-neighbours and mean-field interactions.

## 1. Introduction

Phase transitions are phenomena which bring about qualitative physical changes in the macroscopic behaviour of a system, even if the microscopic forces acting among the constituents of a system remain the same. After Landau phenomenological theory, phase transitions are associated with the phenomenon of spontaneous symmetry-breaking—the symmetry properties of the Hamiltonian describing a system are shared with the physical states accessible to the system at high temperatures, whereas some symmetries of the Hamiltonian are lost by the accessible states at low temperatures. The thermodynamic variable which characterizes the symmetry-breaking phenomenon is the order parameter, vanishing in the most symmetric phase and taking non-vanishing values in the less symmetric phase. However, the theory is not exhaustive because there are several systems undergoing phase transitions that do not fit in this framework: liquid-gas transitions, Kosterlitz-Thouless transitions, transitions in gauge theories, transitions in glasses and supercooled liquids, transitions in amorphous and disordered systems, folding transitions in homopolymers and proteins, all these lack an order parameter. Moreover, phase transition phenomena are experimentally observed in very small systems, at meso and nano scales, what is at odds with the thermodynamic limit dogma stemming from the Yang-Lee theory [1]. Therefore, during a recent past, in the attempt to generalize the existing theories, the focus has been put on the roots of statistical mechanics which has been devised to get rid of microscopic Hamiltonian dynamics under the assumption of its ergodicity. After the Poincaré-Fermi theorem, Hamiltonian systems with a large number of degrees of freedom are generically non-integrable and the only constant of motion is total energy so that the dynamics is *bona fide* ergodic. In addition, when the number of degrees of freedom already exceeds a few tens, all the phase space is filled with chaotic trajectories thus entailing also the mixing property of the dynamics [2]. On the one hand modern computers have made possible numerical testing of the founding assumptions of statistical mechanics, on the other hand have made possible the numerical investigation of phase transitions through Hamiltonian dynamics, that is, in the framework of the microcanonical ensemble. In fact, the ergodic invariant measure of Hamiltonian dynamics is the microcanonical measure [2]. Independently, a Hamiltonian flow can be identified with a geodesic flow on a Riemannian differentiable manifold endowed with a suitable metric (let us call these manifolds *mechanical manifolds*) [2] so that, by combining this fact with the numerical investigation of phase transitions through Hamiltonian dynamics, a natural question arises: what happens to the mechanical manifolds in presence of a phase transition? After several investigations [3,4,5,6], the answer is that peculiar geometrical changes of the energy level sets foliating the phase space occur in correspondence of a phase transition. More specifically, these geometrical changes have been found to stem from deeper topological changes of the configuration space submanifolds defined by $V_{N}\left(q_{1},\ldots ,q_{N}\right)=v\in R$ where $V(q_{1},\ldots ,q_{N})$ is the potential function; these level sets are equivalently denoted by $\Sigma_{v}^{V_{N}}=V_{N}^{-1}\left(v\right)$. Whence the *Topological Hypothesis* on the deep origin of phase transitions and on the loss of analyticity of thermodynamic observables in the N→∞ limit. This hypothesis has been rigorously proved in the case of a few exactly solvable models [2].

Afterwards, it has been proved that for a large class of physical systems, a phase transition *necessarily* stems from the loss of diffeomorphicity of the $M_{v}^{V_{N}}=V_{N}^{-1}\left(\left(-\infty ,v\right]\right)$, and, equivalently, of the $\Sigma_{v}^{V_{N}}$ [7,8,9]: diffeomorphicity among the members of the family ${\left\{M_{v}^{V_{N}}\right\}}_{v\in R}$, for any *N* larger than some $N_{0}$, implies the absence of phase transitions. On this basis it has been shown that phase transitions necessarily stem from topological part of an analytical expression relating thermodynamic entropy with some topological invariants of the mentioned submanifolds of phase space [2,9]. Topological concepts entered the study of phase transitions in a wide variety of problems, for example, in the study of entropy-driven transitions [10,11] (having even applications to robotics), in the field of quantum phase transitions [12,13,14,15], or in that of gases and supercooled liquids [16,17], in classical models in statistical mechanics [18,19,20], and to study DNA denaturation [21], peptide structure [22], among the others. Besides that, the proposed approach is of prospective interest to tackle transitions in: *(i)* finite/small *N* systems; *(ii)* the microcanonical ensemble; *(iii)* systems without an order parameter; *(iv)* complex network systems.

Let us concisely recall that the microcanonical ensemble provides the statistical description of an isolated system at equilibrium. Within this ensemble thermodynamic quantities, like temperature and specific heat, are derived from the entropy which is defined, according to Boltzmann, [23] as
(1)$$S\left(E,\alpha \right)=k_{B}\ln\left(\omega \left(E,\alpha \right)\Delta \right)\;,$$
where $k_{B}$ is the Boltzmann constant, ω(E,α) is the density of microstates per unit energy interval, *E* is the total energy, $\alpha =(\alpha_{1},\ldots \alpha_{m})$ summarizes external parameters as, for instance, the volume, and Δ is a constant with the dimension of an energy. In the following we assume units such that $k_{B}=1$. It is not out of place to mention that the above given definition has recently been, and to some extent still is controversial, in fact it has been recently argued [24] that only the Gibbs definition of entropy yields a consistent thermodynamics, whereas this would not be the case of Boltzmann entropy. To the contrary, we have pointed out in References [13,14,25] that the Boltzmann definition of entropy actually provides a consistent thermodynamics.

Entropy S(E,α) constitutes the fundamental thermodynamic potential in the microcanonical ensemble. Indeed, from the entropy *S* of a given system, secondary thermodynamic observables, such as absolute temperature *T* or pressure *p* are derived by differentiation of *S* with respect to the parameters {E,α}. Let us denote the partial derivatives with respect to *E* by a prime, in this case temperature *T* is given by
(2)$$T={\left(\frac{\partial S}{\partial E}\right)}^{-1}=\frac{\omega }{\omega^{\prime }}\;.$$
Similarly, for the specific heat we have
(3)$$C_{v}={\left(\frac{\partial T}{\partial E}\right)}^{-1}=-\frac{{\left(\frac{\partial S}{\partial E}\right)}^{2}}{\left(\frac{\partial^{2}S}{\partial E^{2}}\right)}=\frac{{\left(\omega^{\prime }/\omega \right)}^{2}}{{\left(\omega^{\prime }/\omega \right)}^{2}-\omega^{\prime \prime }/\omega }\;.$$

In the following, we will derive the explicit formulas for several thermodynamic quantities in the case of a generic Hamiltonian system. Indeed, in this case, the geometric structure related to the dynamics allows one to derive explicit formulas that can be used in numerical simulations of a Hamiltonian flow to work out thermodynamic observables through time averages. Furthermore, we consider two lattice models for which we specify such formulas and for which we perform numerical simulations in order to exemplify the method. In addition, we consider several geometric quantities that seem to clearly detect the phase transition point.

## 2. Geometric Microcanonical Thermodynamics

Consider a generic classical many-particle system described by an autonomous Hamiltonian $H(x_{1},\ldots ,x_{N})$ depending on *N* canonical coordinates, x=(p,q) in which the energy is the only first integral of motion. In this case the Boltzmann entropy reads
(4)$$S\left(E\right)=\ln\;\int \;d^{N}x\;\delta \left(E-H\left(x\right)\right)\;,$$
where δ is the Dirac function.

In accordance with the conservation of energy, during its evolution in time the representative point of the system moves on a given energy-level set. The Liouville theorem shows that the measure of the Euclidean volume is preserved by the dynamics. Consistently, the invariant measure μ induced on each energy level set $\Sigma_{E}$, of energy *E*, is given by [26]
(5)$$d\mu =\frac{d\Sigma }{∥\nabla H∥}\;,$$
where dΣ is the Euclidean measure induced on $\Sigma_{E}$, and ∥·∥ is the Euclidean norm. Remarkably, the density of microstates corresponding to *E* just depends on such invariant measure as
(6)$$\omega \left(E\right)=\int_{\Sigma_{E}}d\mu \;.$$
For any function Φ of the canonical coordinates the average 〈Φ〉 is given by
(7)$$\langle \Phi \rangle =\frac{\int_{\Sigma_{E}}d\mu \;\Phi }{\int_{\Sigma_{E}}d\mu }\;.$$
For special choices of Φ one obtains thermodynamical observables, like temperature, order parameter, specific heat and so on, as averaged combinations of the derivatives of the microcanonical entropy with respect to the energy.

This has been used by Rugh in Referene [27] for Hamiltonian systems for which the energy is the only conserved quantity and, in Reference [28] and Reference [29] for the case of two or more conserved quantities, respectively. The formalism derived in [28] has been successfully adopted for the study of the microcanonical thermodynamics of systems describing Bose-Einstein condensates in optical lattices [13,14,30,31] for which there exist two conserved quantities.

The two $\phi^{4}$ models considered in the following have only one conserved quantity, the total energy, therefore from now on we will limit our discussion to the the simpler case studied in Reference [27]. The geometric key tool in this case is the Federer-Laurence derivation formula [32,33]
(8)$$\frac{\partial^{k}}{\partial E^{k}}\left(\int_{\Sigma_{E}}\psi d\Sigma \right)=\int_{\Sigma_{E}}A^{k}\left(\psi \right)d\Sigma \;,$$
where
(9)$$A\left(•\right)=\frac{1}{∥\nabla H∥}\nabla \left(\frac{\nabla H}{∥\nabla H∥}•\right)\;.$$

By using this formula in the inverse temperature definition
(10)$$\frac{1}{T}=\frac{\partial S}{\partial E}$$
one obtains
(11)$$\beta =\frac{1}{T}=\frac{\partial }{\partial E}\ln\left(\omega \left(E,\alpha \right)\Delta \right)=\frac{\omega^{\prime }}{\omega }=\langle \Phi_{1}\rangle \;,$$
where $\omega^{\prime }=\frac{\partial \omega }{\partial E}$ and using Equation (Equation 8)
(12)$$\langle \Phi_{1}\rangle =\frac{\partial }{\partial E}\left(\ln\int_{\Sigma_{E}}\frac{1}{∥\nabla H∥}d\Sigma \right)={\left(\int_{\Sigma_{E}}\frac{1}{∥\nabla H∥}d\Sigma \right)}^{-1}\int_{\Sigma_{E}}\frac{1}{∥\nabla H∥}\nabla \cdot \left(\nabla H/{∥\nabla H∥}^{2}\right)d\Sigma \;.$$

It is worth mentioning here that in a recent paper [25], one of the present authors has suggested using the surface entropy, that is the logarithm of the area of the constant energy hypersurfaces in the phase space, as the definition for the thermodynamic microcanonical entropy, in place of the standard definition (Equation 4). Besides the fact that the surface entropy has properties which make it an attractive definition for small systems [34], from a geometric point of view, the inverse temperature $\beta_{s}$ derived from the surface entropy is linked to the mean curvature of the hypersurface H(x)=E, that is with a geometric quantity. In fact, in Reference [34] it is shown that in the case of the surface entropy the inverse temperature results
(13)$$\beta_{s}=\frac{\int_{\Sigma_{E}}M\left(x\right)m^{N-1}\left(\Sigma_{E}\right)}{\int_{\Sigma_{E}}m^{N-1}\left(\Sigma_{E}\right)}\;,$$
where
(14)$$M\left(x\right)=\frac{1}{∥\nabla H∥}\nabla \cdot \left(\frac{\nabla H}{∥\nabla H∥}\right)$$
is the local mean curvature divided by ∥∇H∥.

Coming back to the standard entropy definition (Equation 4), after the Federer-Laurence derivation formula (Equation 8) we get
(15)$$\frac{\omega^{\prime \prime }}{\omega }=\langle \Phi_{2}\rangle \;,$$
where
(16)$$\Phi_{2}={\nabla \cdot (\nabla H/∥\nabla H∥}^{2}\Phi_{1})\;,$$
thus the specific heat (Equation 3) results
(17)$$C_{v}=\frac{{\langle \Phi_{1}\rangle }^{2}}{{\langle \Phi_{1}\rangle }^{2}-\langle \Phi_{2}\rangle }\;.$$

More generally, the derivative of order *k* is obtained by a recursion according to the relation
(18)$$\frac{1}{\omega }\frac{\partial^{k}\omega }{\partial E^{k}}=\langle \Phi_{k}\rangle$$
where
(19)$$\Phi_{k}={\nabla \cdot (\nabla H/∥\nabla H∥}^{2}\Phi_{k-1})\;.$$
We will discuss in the following the relevance of the behaviour of the second derivative of the entropy with respect to the energy density E/N. The latter quantity is expressed in terms of the averages of $\Phi_{1}$ and $\Phi_{2}$ according to the following equation
(20)$$\left(\frac{\partial^{2}S}{\partial E^{2}}\right)=\langle \Phi_{2}\rangle -{\langle \Phi_{1}\rangle }^{2}$$

In the following we will report the microcanonical inverse temperature β(E) and the specific heat $C_{v}\left(E\right)$ by time averages of the relevant functions $\Phi_{1}$ and $\Phi_{2}$. In fact, under the hypothesis of ergodicity, the microcanonical averages of each observable Φ can be equivalently measured along the dynamics according to
(21)$$\langle \Phi \rangle =\lim_{\tau \to \infty }\frac{1}{\tau }\int_{0}^{\tau }dt\;\Phi \left(t\right)\;,$$
equivalent to an average on $\Sigma_{E}$ as in Equation (Equation 7).

The explicit form for the function $\Phi_{1}$ is
(22)$$\Phi_{1}=\frac{\;△H}{{∥\nabla H∥}^{2}}-2\frac{\nabla H\cdot H\cdot \nabla H}{{∥\nabla H∥}^{4}}\;,$$
where H is the Hessian matrix of the Hamiltonian function, whereas $\Phi_{2}$ is a little bit more complicated
(23)$$\Phi_{2}=\Phi_{1}^{2}+\frac{\nabla H\cdot \nabla (\Phi_{1})}{{∥\nabla H∥}^{2}}\;.$$

### Phase Transitions in the Microcanonical Ensemble

The inequivalence of statistical ensembles in presence of long-range interactions, and Molecular Dynamics studies of energy conserving systems have motivated several investigations of the microcanonical description of phase transitions [35,36,37,38,39,40]. In particular, we emphasize a recent and very interesting proposal in Reference [40] which proves very effective to interpret the outcomes of numerical simulations, as it will be seen in the following. A complementary viewpoint *à la Ehrenfest* has been heuristically put forward in Reference [41]. This proceeds from the fact that the natural counterpart of microscopic Hamiltonian dynamics is the microcanonical ensemble where, as we have already recalled above, the relevant thermodynamic potential is entropy. From the latter, one derives the specific heat (Equation 3), where we see that $C_{v}$ can diverge only as a consequence of the vanishing of $\left(\partial^{2}S/\partial E^{2}\right)$ which a-priori has nothing to do with a loss of analyticity of S(E). This disagrees with Ehrenfest’s classification of phase transitions in the canonical ensemble, associated with a loss of analyticity of Helmholtz free energy, and thus also of the entropy. As is well known, the identification of a phase transition with an analyticity loss of a thermodynamic potential (in the gran-canonical ensemble) is rigorously stated by the Yang-Lee theorem. Ehrenfest’s classification of phase transitions in the canonical ensemble is based on the order of the derivative of free energy which is singular at the transitions point, but this way of classification turned out inadequate after Onsager’s exact solution of the 2D Ising model for which the specific heat has been found divergent, thus entailing a discontinuity of the first order derivative of free energy. Therefore, the distinction between first and second order phase transitions is lost.

Coming to the microcanonical ensemble, for standard Hamiltonian systems (i.e., quadratic in the momenta) the relevant information is carried by the configurational microcanonical entropy
$$S_{n}\left(\bar{v}\right)=\frac{1}{n}\log\int dq_{1}\cdots dq_{n}\;\delta \left[V_{n}\left(q_{1},\cdots ,q_{n}\right)-v\right]\;,\;$$
where $\bar{v}=v/n$ is the potential energy per degree of freedom, δ[·] is the Dirac function, $S_{n}\left(\bar{v}\right)$ is related to the configurational canonical free energy
$$f_{n}\left(\beta \right)=\frac{1}{n}\log\int dq_{1}\ldots dq_{n}\;e^{-\beta V_{n}\left(q_{1},\ldots ,q_{n}\right)}$$
for any n∈N, $\bar{v}\in R$, and β∈R, through the Legendre transform
(24)$$\begin{array}{c} -f_{n}\left(\beta \right)=\beta \cdot {\bar{v}}_{n}-S_{n}\left({\bar{v}}_{n}\right)\;, \end{array}$$
where the inverse of the configurational temperature T(v) is given by $\beta_{N}\left(\bar{v}\right)=\partial S_{N}\left(\bar{v}\right)/\partial \bar{v}$.

Then consider the function $\phi \left(\bar{v}\right)=f_{n}\left[\beta \left(\bar{v}\right)\right]$, from $\phi^{\prime }\left(\bar{v}\right)=-\bar{v}\;\left[d\beta_{n}\left(\bar{v}\right)/d\bar{v}\right]$ we see that if $\beta_{n}\left(\bar{v}\right)\in C^{k}\left(R\right)$ then also $\phi \left(\bar{v}\right)\in C^{k}\left(R\right)$ which in turn means $S_{n}\left(\bar{v}\right)\in C^{k+1}\left(R\right)$ while $f_{n}\left(\beta \right)\in C^{k}\left(R\right)$. Hence, if the functions ${\left\{S_{n}\left(\bar{v}\right)\right\}}_{n\in N}$ are convex, thus ensuring the existence of the above Legendre transform, and if in the n→∞ limit it is $f_{\infty }\left(\beta \right)\in C^{0}\left(R\right)$ then $S_{\infty }\left(\bar{v}\right)\in C^{1}\left(R\right)$, and if $f_{\infty }\left(\beta \right)\in C^{1}\left(R\right)$ then $S_{\infty }\left(\bar{v}\right)\in C^{2}\left(R\right)$. So far we have seen that, generically (that is apart from any possible counterexample), if $f_{n}\left(\beta \right)\in C^{k}\left(R\right)$ then $S_{n}\left(\bar{v}\right)\in C^{k+1}\left(R\right)$. This all what is needed to *heuristically* proceed to a classification of phase transitions *à la* Ehrenfest in the present microcanonical configurational context. By analogy with the original Ehrenfest’s definition associating a first or second-order phase transition with a discontinuity in the first or second derivatives of $f_{\infty }\left(\beta \right)$, respectively, we associate a first (second) order phase transition with a discontinuity of the second (third) derivative of the entropy $S_{\infty }\left(\bar{v}\right)$. It is worth emphasizing that this definition of the order of a phase transition is given regardless of the existence of the Legendre transform. Indeed, the latter is very often not defined in presence of first-order phase transitions which bring about a kink-shaped entropy as a function of the energy [35]. Therefore, rigorously, the definition that we are putting forward does not stem neither mathematically nor logically from the original Ehrenfest classification.

This entropy-based classification of phase-transitions *à la* Ehrenfest, although to some extent arbitrary, has a *heuristic* motivation. Moreover, it does not suffer any longer the difficulty arising in the framework of canonical ensemble stemming from the 2D Ising model as it has been recalled above. This classification is useful also in case of ensemble non-equivalence when only the microcanonical description is the only correct one.

The usefulness of this classification has to be confirmed against practical examples beyond Ref. [41].

## 3. The Models

In what follows two different versions of a $\phi^{4}$ model are considered. These are defined through nearest-neighbours interactions and through long-range interactions, respectively. These models are in some sense “paradigmatic” in what they both undergo a second order phase transition due to the $Z_{2}$ symmetry-breaking, the same of the 2D Ising model.

The $\phi^{4}$ models are defined by the Hamiltonian
(25)$$H=\sum_{j}\frac{1}{2}\pi_{j}^{2}+V\left(\phi \right)$$
where
(26)$$\;\;V\left(\phi \right)=\sum_{j}\left[\frac{\lambda }{4!}\phi_{j}^{4}-\frac{\mu^{2}}{2}\phi_{j}^{2}+\frac{J}{D}\sum_{k\in I(j)}{\left(\phi_{j}-\phi_{k}\right)}^{2}\right]\;,$$
$\pi_{j}$ is the conjugate momentum of the variable $\phi_{j}$ that defines the position of the $j^{th}$ particle. In the case of the two dimensional model, $j=(j_{1},j_{2})$ denotes a site of a two dimensional lattice, the number of nearest neighbours is D=4 and I(j) are the nearest neighbour lattice sites of the $j^{th}$ site. The coordinates of the sites are integer numbers $j_{k}=1,\ldots ,N_{k}$, k=1,2, so that the total number of sites in the lattice is $N=N_{1}\;N_{2}$. Furthermore periodic boundary conditions are assumed. In the case of the mean-field model j=1,…,N denotes the indices of the 2N canonical coordinates of the system, D=N−1 and I(j)=1,…,N. The Hamiltonian equations of motion read
(27)$$\begin{array}{c} {\dot{\phi }}_{j}=\pi_{j}\;,\\ {\dot{\pi }}_{j}=-\frac{\partial V}{\partial \phi_{j}} \end{array}\;.$$
The local potential displays a double-well shape whose minima are located at $\pm \sqrt{3!\mu^{2}/\lambda }$ and to which it corresponds the ground-state energy per particle $e_{0}=-3!\mu^{4}/\left(2\lambda \right)$. At low-energies the system is dominated by an ordered phase where the time averages of the local fields are not vanishing. By increasing the system energy the local $Z_{2}$ symmetry is restored and the averages of the local- fields are zero.

Naturally, the explicit form for the geometric quantities entering in
(28)$$\Phi_{1}=\frac{\;△H}{{∥\nabla H∥}^{2}}-2\frac{\nabla H\cdot H\cdot \nabla H}{{∥\nabla H∥}^{4}}\;,$$
and
(29)$$\Phi_{2}=\Phi_{1}^{2}+\frac{\nabla H}{{∥\nabla H∥}^{2}}\cdot \nabla \left(\Phi_{1}\right)\;,$$
where
(30)$$\begin{array}{c} \frac{\nabla H}{{∥\nabla H∥}^{2}}\cdot \nabla (\Phi_{1})=\\ \sum_{jk}\frac{\partial_{j}H\partial_{jkk}^{3}H}{{∥\nabla H∥}^{4}}-2\sum_{jkr}\frac{\partial_{j}H\partial_{k}H\partial_{r}H\partial_{jkr}^{3}H}{{∥\nabla H∥}^{6}}+\\ -4\frac{\nabla H\cdot H\cdot H\cdot \nabla H}{{∥\nabla H∥}^{6}}-2\frac{\nabla H\cdot H\cdot \nabla H}{{∥\nabla H∥}^{4}}\times \\ \left(\Phi_{1}-2\frac{\nabla H\cdot H\cdot \nabla H}{{∥\nabla H∥}^{4}}\right)\;, \end{array}$$
depend on the details of the Hamiltonian of each model. Therefore, in the following we will consider these two cases separately and we will set
$$\nabla \equiv \left(\begin{array}{c} \vdots \\ \partial_{\pi_{j}}\\ \vdots \\ \partial_{\phi_{j}}\\ \vdots \end{array}\right)\;.$$

### 3.1. 2-d $\phi^{4}$ Model

In the case of the two dimensional model we have
(31)$$\;△H=N\left(1+4J-\mu^{2}\right)+\frac{\lambda }{2!}{∥\phi ∥}^{2}\;,$$
where $∥\phi ∥=\sqrt{\sum_{j}\phi_{j}^{2}}$. In addition it results
(32)$$∥\nabla H∥=\sqrt{2K+{∥\nabla V∥}^{2}}\;,$$
where *K* stands for the total kinetic energy $K=\sum_{j}\pi_{j}^{2}/2$ and
(33)$$\nabla_{k}V=\frac{\lambda }{3!}\phi_{k}^{3}+\left(4J-\mu^{2}\right)\phi_{k}-J\sum_{j\in I(k)}\phi_{j}\;.$$
The Hessian matrix of the Hamiltonian function is
(34)$$H=\left(\begin{array}{cc} I & 0\\ 0 & H_{V} \end{array}\right)\;,$$
where the entries of the Hessian matrix $H_{V}$ of the potential function *V* result
$${\left(H_{V}\right)}_{ij}=\partial_{ij}^{2}V=\left(\frac{\lambda }{2!}\phi_{j}^{2}+4J-\mu^{2}\right)\delta_{i,j}-J\delta_{j,I(i)}\;.$$
Finally, it is
$$\partial_{ijk}^{3}V=\lambda \delta_{i,j}\delta_{j,k}\phi_{j}\;.$$

### 3.2. Mean-Field $\phi^{4}$ Model

The analogous quantities for the case of the mean-field model are the following. △H has the same form of (Equation 31), whereas
(35)$$\nabla_{k}V=\frac{\lambda }{3!}\phi_{k}^{3}+\left[4J\frac{N}{N-1}-\mu^{2}\right]\phi_{k}-\frac{4J}{N-1}M\;,$$
where we have introduced the total magnetization
(36)$$M=|\sum_{j}\phi_{j}|\;.$$
In this case the Hessian matrix $H_{V}$ of the potential function *V* is
$${\left(H_{V}\right)}_{ij}=\partial_{ij}^{2}V=\left[\frac{\lambda }{2!}\phi_{j}^{2}+4J\frac{N}{N-1}-\mu^{2}\right]\delta_{i,j}-\frac{4J}{N-1}\;,$$
and $\partial_{ijk}^{3}V$ has the same form of the 2−d case.

## 4. Numerical Results

We have investigated the microcanical thermodynamics of these two systems by measuring some geometric quantities as illustrated above which are relevant to catch the thermodynamical properties of these models at their respective phase transition points. Thus, we have numerically integrated the equations of motion (Equation 27) of both models, by using a third order symplectic algorithm [42] and starting from random initial conditions corresponding to different values of the total energy *E*. In such a way, we have measured—along the dynamics—the time averages of the quantities $\Phi_{1}$ and $\Phi_{2}$ for several values of the total energy *E*, according to Equation (Equation 21). From the time averages $\langle \Phi_{1}\rangle \left(E\right)$ and $\langle \Phi_{2}\rangle \left(E\right)$ by means of Equations (Equation 11) and (Equation 17), we have derived the caloric curve T(E) and the specific heat $C_{v}\left(E\right)$ of the two models. In addition to the thermodynamic quantities, we have measured geometric quantities as the average of the Ricci curvature $K_{R}\left(q,\dot{q}\right)$ (see Appendix A for details). The main outcome of our analysis is the better effectiveness of the geometric indicators as phase-transitions detectors with respect to the traditional thermodynamic indicators, with the exception of the order parameter. In a recent paper [41], by resorting to geometric indicators, it has been possible to unambiguously characterize and explain the phenomenology of a system that undergoes a thermodynamic phase transition in the absence of a global symmetry-breaking and thus in the absence of an order parameter.

### 4.1. 2-d $\phi^{4}$ Model

In this section, we report the results of the simulations performed for the 2d$\phi^{4}$ model (with nearest-neighbour interactions). The order parameter M=〈M〉/N—average of the total magnetization M defined in (Equation 36)—is reported as a function of the energy density ϵ=E/N in Figure 1: the bifurcation pattern of M(E/N) is typical of a second-order phase transition.

Figure 1 allows one to determine the critical energy density $\epsilon_{c}$ of the phase-transition, which is found to be $\epsilon_{c}\approx 11.1$.

Another typical signature of a phase transition is provided by the shape of the caloric curve T(E), that is. the temperature as a function of the energy. In the case of the 2d$\phi^{4}$ model, we have derived such a curve by time-averaging $\Phi_{1}$, along with the dynamics, for different initial conditions corresponding to several energy densities. The caloric curve derived by $1/\langle \Phi_{1}\rangle$, according to Equation (Equation 11), is reported in Figure 2. In the case of the 2d$\phi^{4}$ model, the caloric curve T(E/N) displays an inflection point just at the critical energy density value identified by the bifurcation point of the order parameter—highlighted with the vertical dfashed line in Figure 2—and this is in perfect agreement with the proposition put forward by Bachmann in Refs. [40,43].

Through time averages of $\Phi_{1}$ computed along with the numeric integration of the equations of motion for different initial conditions, we have derived the curve of the inverse temperature β as a function of E/N. Figure 3 shows this curve for the 2d$\phi^{4}$ model. Also in the case of β(E/N), the transition point $E_{c}/N=\epsilon_{c}$ (located by the dashed vertical line in the same figure) corresponds to an inflection point of this curve.

The expected growth with the system-size, of the peak of the specific-heat in correspondence of the phase-transition is shown in Figure 4. The curve of the specific heat $C_{V}$ vs the energy-density E/N has been obtained via Equation (Equation 17) where the averages have been again computed by means of time averages of the quantities (22) and (Equation 23) according to Equation (Equation 21), for different lattice sizes, that is, 24×24 sites (open circles), 48×48 sites (open squares) and 128×128 sites (crosses).

Figure 5 reports the second derivative of the entropy with respect to the energy *E*. As mentioned above, the divergence of the specific heat stems from the vanishing of this derivative. This figure displays the outcomes of a numerical derivation of the curve β(E) obtained for systems of different sizes: 24×24 lattice sites (open circles), and 48×48 lattice sites (crosses). In addition, Figure 5 reports the values of $N\partial^{2}S/\partial E^{2}$ vs E/N derived by means Equation (Equation 20) through time averages of $\Phi_{1}$ and $\Phi_{2}$ in the case of a system with 24×24 (open squares), 48×48 (full circles) and 128×128 (stars) lattice sizes. The figure shows distinctly the transition point, corresponding to a discontinuity of the fourth order of the derivative of *S*. Remarkably, these numerical outcomes confirm that the growth with the system-size of the specific-heat, in correspondence of the phase-transition, as a consequence of the approaching of $N\partial^{2}S/\partial E^{2}$ to zero as per Equation (Equation 3). Figure 5 shows that the larger the system size the closer the value of $N\partial^{2}S/\partial E^{2}$ to zero, in correspondence of the phase-transition point.

In Figure 6 the curve 〈▵H〉/N vs E/N is reported, that is the time average of the Laplacian of the Hamiltonian function per degree of freedom, and again it clearly shows an inflection point at the transition energy density. The quantity 〈▵H〉/N has a geometric meaning but of a different kind with respect to those related with the extrinsic curvature of the energy level sets. In fact, as shown in the Appendix A, it turns out that the Laplacian of the Hamiltonian [in Equation (Equation A15)] coincides, apart from a constant, with the Ricci-curvature of a Riemannian manifold, an enlarged configurational space-time endowed with a metric due to Eisenhart [2,44]. The geodesics of this manifold are just the natural motions of the Newton equations associated with the Hamiltonian of the system.

### 4.2. Mean-Field $\phi^{4}$ Model

In the present section, we report the results of the numerical simulations performed for the mean-field $\phi^{4}$ model. Also this model undergoes a second-order phase transition which is clearly displayed by the bifurcation of the order parameter M=〈M〉/N, the magnetization, versus the energy density ϵ=E/N, as is shown in Figure 7 where the critical energy density of the phase transition point is found to be $\epsilon_{c}\approx 25$.

With respect to the 2d model, the long-range interactions make this system harder to simulate. In fact, considerable difficulties have been encountered in computing stabilized time averages of the same quantities computed for the $\phi^{4}$ model with short-range interactions. These difficulties depend on the worsening of the properties of self-averaging of this model for energy values close to the transition point, clearly due the long-range interactions. Besides that, and again except for the order parameter, the mean-field model undergoes a phase transition which appears much "softer" than the one undergone by the 2d model. This fact is put in evidence by the basic thermodynamic functions T(E/N) and β(E/N), computed though the time averages of $\Phi_{1}$ along with the numeric integration of the equations of motion for different initial conditions, and reported in Figure 8 and Figure 9, respectively. In particular the curve β(E/N) does not display at all any feature to identify the presence of a transition. All in all, these functions are not very helpful neither to clearly identify the presence of a phase-transition nor, possibly, its transition point.

In Figure 10 we report the derivative $N\partial^{2}S/\partial E^{2}$ as a function of E/N worked out in the same way as previously done for the short-range model. The energy density pattern of this derivative is found to be very noisy, even after many millions of integration time steps, and this goes together with a very bad outcome for the specific heat, which, on purpose, is not reported here. To the contrary, and together with the order parameter, Figure 11 shows an interesting pattern of the time average of the Ricci curvature of the mechanical manifold $\left(M\times R^{2},g_{e}\right)$ (see Appendix A) as a function of the energy density. The pattern of 〈△H〉(E/N)/N displays a “cuspy” point in correspondence with the vertical red dashed line locating the phase transition point. Of course, within the obvious limits of numerical outcomes, such a “cuspy” point appears as an abrupt change of the second derivative of the Ricci curvature—with respect to the energy—because above the transition point its pattern appears convex (of positive second derivative), whereas just below the transition point the values of the Ricci curvature appear to align along a straight segment, thus with a vanishing second derivative. Loosely speaking, this is reminiscent of similar jumps of the second derivative with respect to the energy of an average curvature function which has been found for a gauge model [41].

## 5. Concluding Remarks

We have considered the second order phase transitions stemming from the same kind of $Z_{2}$ symmetry-breaking phenomenon occurring in two $\phi^{4}$ models. Besides the standard detection of the presence of a phase transition through the bifurcation of an order parameter, we have focused on basic geometric properties of different manifolds, highlighting that the values of thermodynamic observables, like temperature and specific heat, and their functional dependence on the energy are the consequences of more fundamental changes with energy of curvature properties of the energy level sets in phase space. The conceptual interest of this fact is that a phase transition phenomenon can be seen as just depending on the interaction potential of the forces acting among the degrees of freedom of a system, that is, the possibility for a system of undergoing a phase transition is already "encoded" in its Hamiltonian function and thus can be read in the variation of some extrinsic curvature properties of the hypersurfaces H(p,q)=E foliating the phase space. When the variations with energy the geometry of these level-set manifolds are too "mild", as is the case of the mean-field $\phi^{4}$ model, one can again recover a rather sharp geometric signature of the transition by considering the energy variation of the Ricci curvature of a manifold the geodesics of which are the motions of the system. In other words, in both cases, a phase transition phenomenon can be seen as stemming from a deeper level than the usual one which consist of attributing them to a loss of analyticity of the statistical measures in the thermodynamic limit. The statistical measures represent an "epistemic" description of the occurrence of phase transitions, in what statistical measures do not correspond to physically measurable entities, whereas the forces acting among the degrees of freedom of a system belong to an "ontic" level because forces are real physical entities, velocities of the kinetic energy and potentials can be in principle measured, so that for an energy conserving closed system the quantities entering the relation H(p,q)=E are real physical ones.

Finally, since geometric indicators, like the Ricci curvature, are independent of the order parameter among the other thermodynamic quantities, the proposed geometric analysis can be applied also in the case of systems that undergo phase-transitions in absence of a global symmetry breaking and thus in the absence of an order parameter [41].

## Figures and Tables

**Figure 1 entropy-22-00380-f001:**
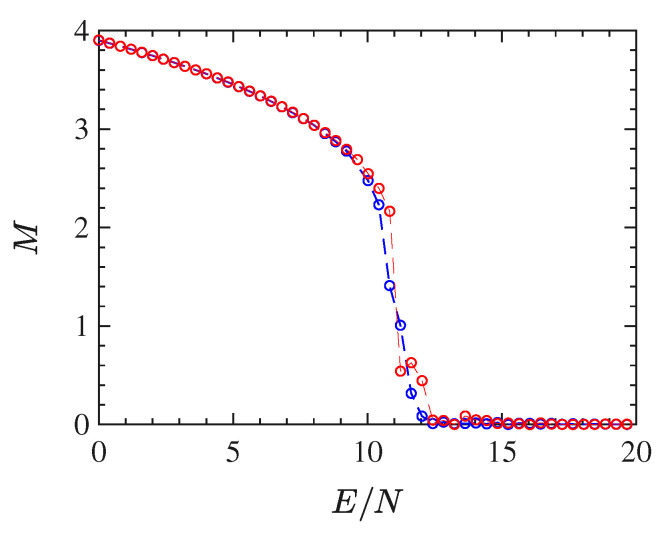
The figure shows the plot of the quantity order parameter *M* vs the energy density E/N for 128×128 particles (blue circles) and 48×48 particles (red circles).

**Figure 2 entropy-22-00380-f002:**
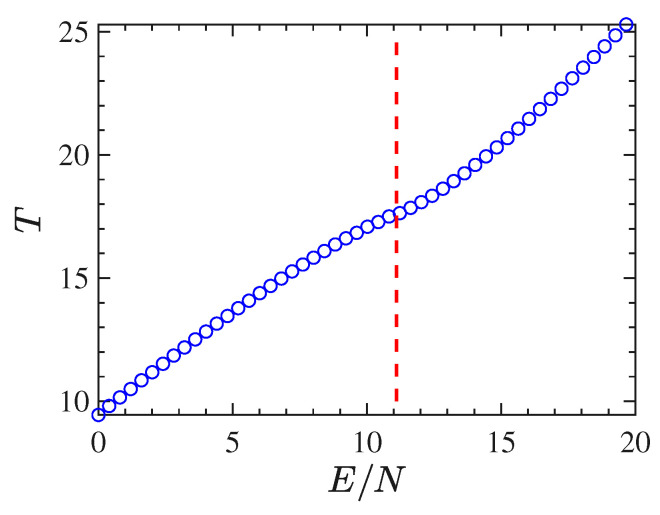
The figure reports the temperature derived by means of time averages of $\Phi_{1}$ whence $T=1/\langle \Phi_{1}\rangle$ according to Equations (Equation 11) and (Equation 21), as a function of the energy density E/N for the 2−d$\phi^{4}$ model for 128×128 particles.

**Figure 3 entropy-22-00380-f003:**
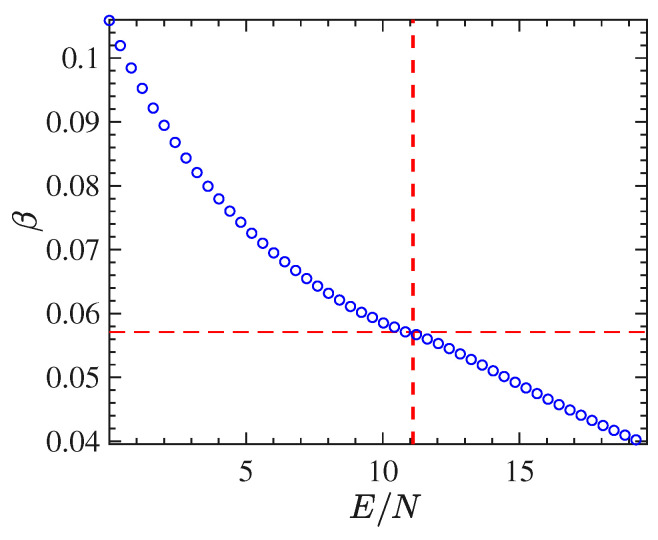
β vs E/N obtained from the time average of $\Phi_{1}$ for several energies *E* in the case of the 2d-$\phi^{4}$ model with 128×128 particles.

**Figure 4 entropy-22-00380-f004:**
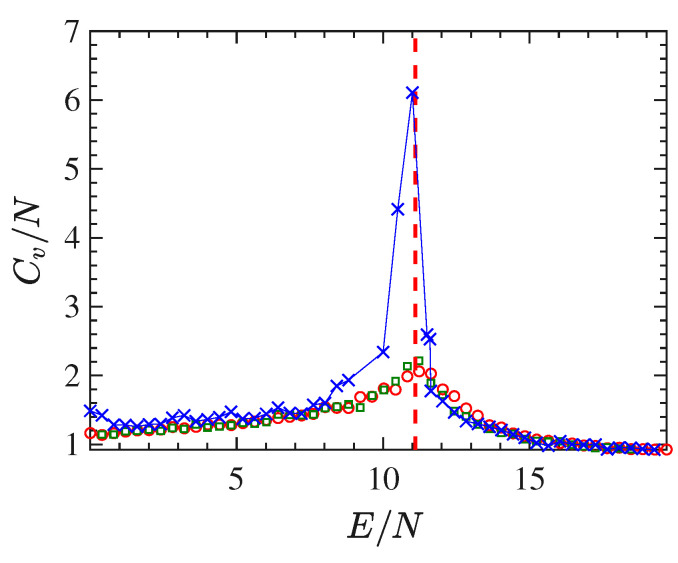
The specific-heat per particle $C_{v}/N$ is reported as a function of the energy density E/N. The lattice sizes are: 24×24 (open circles), 48×48 (open squares) and 128×128 (crosses).

**Figure 5 entropy-22-00380-f005:**
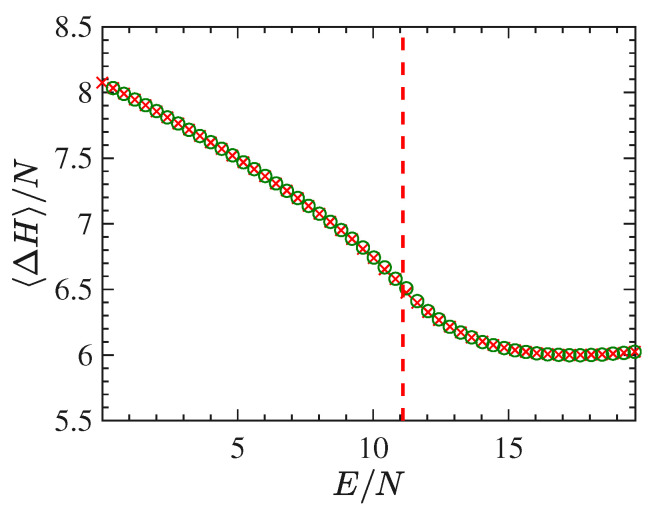
$N\partial^{2}S/\partial E^{2}$ vs E/N derived with a numeric derivative of the curve β(E/N). The latter has been obtained as time average of $\Phi_{1}$ for several values of the total energy *E* in the case of a 24×24 lattice (open circles) and a 48×48 lattice (crosses). Furthermore, the figure plots the $N\partial^{2}S/\partial E^{2}$ derived by the formula $N(\langle \Phi_{2}\rangle -{\langle \Phi_{1}\rangle }^{2})$ in which the averages are temporal. Symbols refer to 24×24 (open squares), 48×48 (full circles) and 128×128 (stars) lattice sizes, respectively. The figure shows distinctly the transition point, corresponding to a discontinuity of the fourth order of the derivative of *S*.

**Figure 6 entropy-22-00380-f006:**
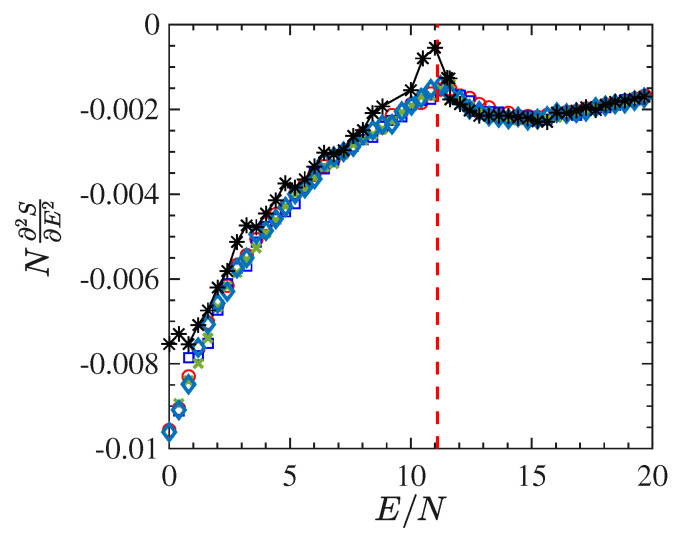
Figure report the time average of △H/N as a function of E/N in the case of a system with 24×24 lattice sites.

**Figure 7 entropy-22-00380-f007:**
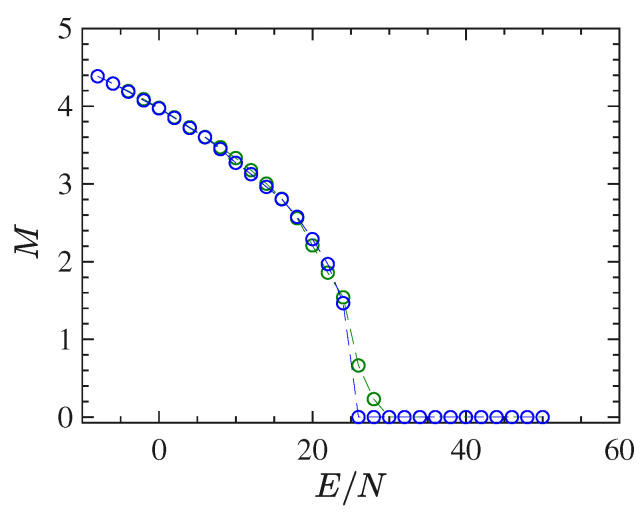
The order parameter *M* for the mean-field $\phi^{4}$ model is reported vs E/N for 1024 particles (green circles) and 2048 particles (blue circles).

**Figure 8 entropy-22-00380-f008:**
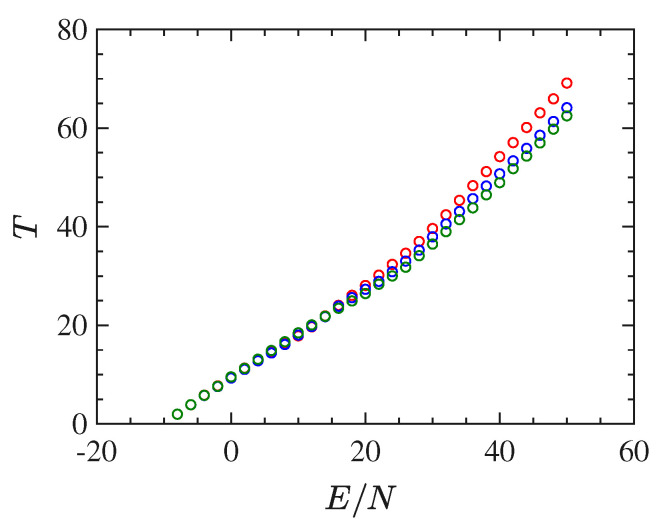
*T* vs E/N for the mean-field $\phi^{4}$ model. N=4096 red circles, N=2048 blue circles, N=1024 green circles.

**Figure 9 entropy-22-00380-f009:**
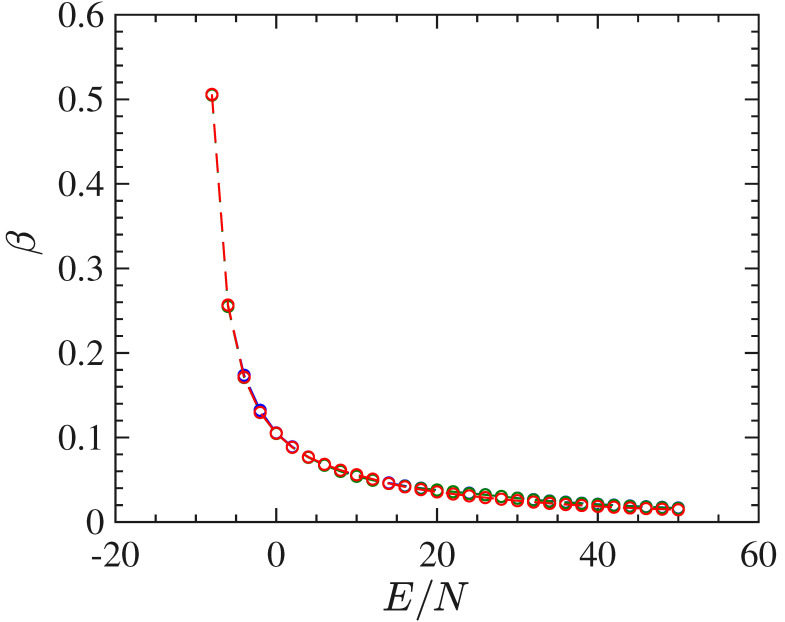
The figure show the curve β vs E/N for the mean-field $\phi^{4}$ model. N=4096 red circles, N=2048 blue circles, N=1024 green circles.

**Figure 10 entropy-22-00380-f010:**
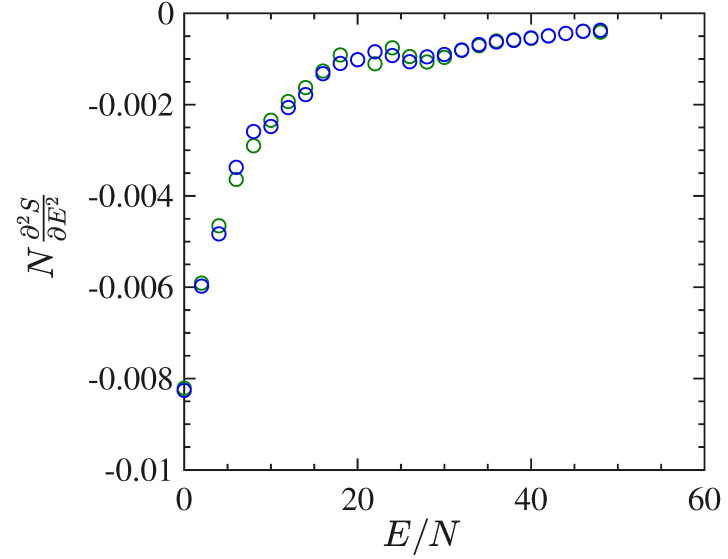
The figure shows the plot of the quantity $N\partial^{2}S/\partial E^{2}$ vs E/N derived with a numeric derivative of the curve β(E) for 1025 particles.

**Figure 11 entropy-22-00380-f011:**
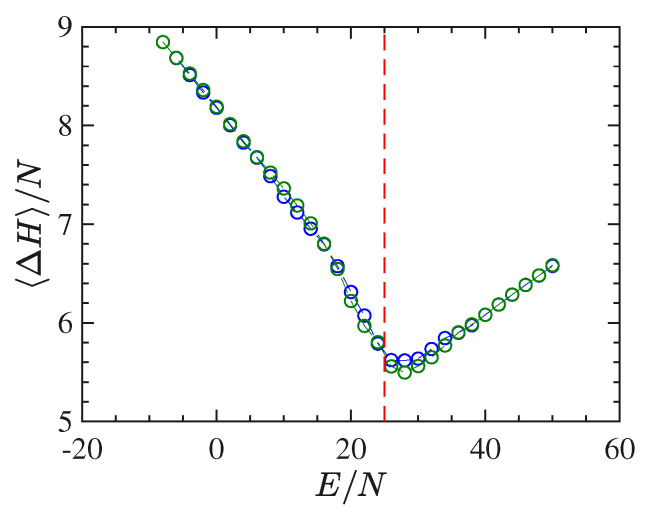
The figure shows the plot of the quantity 〈△H〉/N vs E/N for 1024 particles (green circles) and 2048 particles (blue circles).

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
