# Peer review of "Geometrical Aspects in the Analysis of Microcanonical Phase-Transitions"

_entropy, 2020, doi:10.3390/e22040380_

Round 1

Reviewer 1 Report

The authors should improve the present from

Reviewer 2 Report

In the present manuscript “Geometrical aspects in the analysis of micro canonical phase-transitions” the authors study the connection between phase space geometry and thermodynamic observables in terms of the microcanonical ensemble. Then, within this geometrical point of view, they relate critical behavior to the entropy in their fundamental representation (in terms of energy). By employing this connection, they propose a classification of phase transitions from a microcanonical point of view. Finally, they apply this formalism to phi^4 lattice models with nearest-neighbors and mean-field interactions, by studying numerically the equations of motion under the assumption of ergodicity.

In my opinion the manuscript is interesting, relevant for the current discussion about the definition of entropy in classical and quantum weakly and strong interacting many-body systems, as well as for the equivalence of ensembles, and a promising way to gain a better understanding of phase transitions. Hence, I believe the manuscript is worth for publication in Entropy as a regular article, after the authors address and clarify the following remarks, which I consider important to enhance not only the value and quality of the paper, but also the impact of their results.

1. I consider as the most important problem of the manuscript the lack of discussion about the main advantage of the approach, considering the authors rely numerically on calculating time averages. It is true that grasping thermodynamics from a phase space point of view brings a deeper understanding, but this alone does not undermine the power of the statistical mechanics approach relying on physical motivated order parameters. Then, the authors should emphasize the perks of their approach both for the development of a consistent thermodynamic picture (as they discuss it in the introduction), as well as for the heuristic-motivated Ehrenfest-like classification of phase transitions they propose. In this direction, I suggest the authors to describe first the essentials of the approach with which they are contrasting their results.

2. For other kind of systems, what would be the connection between the heuristic picture of critical behavior in terms of order parameters and the geometrical microcanonical approach proposed by the authors? 

3. It is clear the authors are employing the lattice models as easy to do examples to discuss the formalism. Is there any further motivation to employ these models in particular?

4. The manuscript presents redundant information. For example, lines 99 to 101 in page 8 are almost the same as lines 139 to 142 in page 10. I suggest the authors to describe earlier and with more detail what are the key features distinguishing the nearest neighbors and mean-field model and relevant for the application of their formalism (as they do, for example, in lines 151 to 153 in page 10). Likewise, the caption in Fig. 4 is a repetition of lines 117 to 119 in page 9.

5. I suggest the authors to put together Fig. 1 to Fig. 6 to simplify the reading and make it easier to the visual comparison between the different ways of recognizing the critical behavior including their calculations via time averaging of the observables. The same goes for Figs. 8 to 11.

Finally, the manuscript would benefit of an in deep revision to remove some typos.

Round 2

Reviewer 1 Report

No comments